



# How plant water status drives tree source water partitioning

Magali F. Nehemy[1], Paolo Benettin[2], Mitra Asadollahi[2], Dyan Pratt[1], Andrea Rinaldo[2,3], Jeffrey J. McDonnell[1,4]

[1] Global Institute for Water Security, School of Environment and Sustainability, University of Saskatchewan, 11 Innovation Boulevard, Saskatoon, SK S7N 3H5, Canada
[2] Laboratory of Ecohydrology, Institute of Environmental Engineering, École Polytechnique Fédérale de Lausanne, Station 2, GR C1 575, 1015 Lausanne, Switzerland
[3] Dipartimento ICEA, Università di Padova, via Loredan 20, I-35131 Padova, Italy
[4] School of Geography, Earth & Environmental Sciences, University of Birmingham, Birmingham, UK

*Correspondence to*: Magali Nehemy (magali.nehemy@usask.ca)

**Abstract.** The stable isotopes of oxygen and hydrogen ($\delta^2$H and $\delta^{18}$O) have been widely used to investigate plant water source partitioning. These tracers have shed new light on patterns of plant water use in time and space. However, this black box approach has limited our source water interpretations and mechanistic understanding. Here, we combine measurements of stable isotope composition in xylem and soil water pools with measurements of plant hydraulics, fine root distribution and soil matric potential to investigate mechanism(s) driving tree water source partitioning. We used a 2 m³ lysimeter planted with a small willow tree (*Salix viminalis*) to conduct a high spatial-temporal resolution experiment. We found that tree water source partitioning was driven mainly by tree water status and not by patterns of fine root distribution. Source water partitioning was regulated by plant hydraulic response to changing atmospheric demand and soil matric potential. The depth distribution of soil matric potential appeared to be the largest control on the patterns of soil water partitioning during periods of tree water deficit. Contrary to the common steady state assumption in ecohydrological source water investigations, our results show that tree water use is a dynamic process, driven by tree water deficit. Overall, our findings suggest new research foci for future plant water isotopic investigations, highlighting the importance of hydrometric measurements from the plant perspective.

## 1. Introduction

Tree water use studies using the stable isotopes of hydrogen and oxygen ($\delta^2$H and $\delta^{18}$O) as a tracer have shed considerable light on the ecohydrological processes (Brooks et al., 2010). More recent field-based plant water investigations have shown that plants rarely access percolating rainfall, and use mostly resident, stored soil pore water (Brooks et al., 2010; Goldsmith et al., 2011; Hervé-Fernández et al., 2016). This pool of water stored in soil pores that supplies plant transpiration appears to be isotopically distinct from the mobile rainfall percolating through the soil profile and  contributing  to groundwater recharge and streamflow (Brooks et al., 2010; Sprenger et al., 2019). This ecohydrological separation between plant transpiration and groundwater recharge appears to be widespread across different biomes (Evaristo et al., 2015). However, the mechanisms causing this ecohydrological separation are still unclear (Berry et al., 2017; Penna et al., 2018). At the catchment scale, these mechanisms are vital for a more holistic understanding of the water balance and how the ages of water sampled by roots affect the residence times of subsurface waters and the ultimate streamwater transit times.





Recent investigations have shown that enhanced temporal and spatial sampling resolution of stable isotopes of hydrogen and oxygen is key to improved understanding tree water use patterns linked to soil water dynamics (Allen et al. 2018; Goldsmith et al. 2018; Sprenger et al. 2019). This is crucial because the "age" (transit times) of plant

transpiration can be  many days to months older than the more mobile water that recharge groundwater (Evaristo et al., 2019). Soil water isotopic composition is also spatially heterogeneous, and highly variable (Goldsmith et al., 2018). Thus, explicitly accessing spatial-temporal variation in the stable isotopes of water and root water uptake is key to improve understanding of the mechanisms behind plant water use.

The original conceptual model describing plant use of more tightly bound water pool resident in small soil pore

space was based on the anomalously light rainfall signatures at the time of wet season onset (Brooks et al., 2010). This water storage tightly bound to soil matrix in small pores space is emptied by tree transpiration as the more mobile water becomes unavailable (Brooks et al., 2010). Recent data from other sites supports this complex soil water age dynamics by showing recharge of small pores in the beginning of wet season, and temporal hydrological segregation between small and large soil pores (Sprenger et al., 2019; Xu et al., 2019). But despite the recent

identification of some key issues to be resolved (Berry et al., 2017; Dubbert et al., 2019; Penna et al., 2018), the patterns and mechanisms governing plant water source apportionment are still unclear. And while soil water conditions have been the guiding criteria for sampling campaigns and interpretations about ecohydrological separation (Evaristo et al., 2016; Hervé-Fernández et al., 2016), no studies that we are aware of in this regard, have yet considered plant water status.

Plant water status reflects the plants' response to soil-water supply and atmospheric demand driven by water potential gradients, highly regulated by plant traits, to maintain favorable water balance between pools and fluxes (Fu and Meinzer, 2018; Tilman 1982; Hsiao and Acevedo, 1974). Thus, plant water status provides key information about plant hydraulic functioning and response to environment (Jones, 2007; Steppe, 2018). Fine resolution tree stem diameter change is often used to assess plant water status and understand hydraulic response to external driving

forces  (Dietrich et al., 2018; Steppe et al., 2006; Zweifel et al., 2001; Simonneau et al., 1993; Klepper et al., 1971). Thus, tree stem diameter variability provides an overall indicator of the plant response to surface tensions generated at air- soil-water interface, as the volume of living cells (Nobel, 2009), storage tissues and xylem conduits (Irvine and Grace, 1997; Mencuccini et al. 2017) change according to pressure that they are subjected to. Differences in water potential induces transport of water from high potential to low potential, analogous to Ohm's Law (Larcher

2003). This reflects the movement of water from soil to roots, as well as internal radial transport of water between inner bark and xylem in trees. Plant water status is used often to achieve precision irrigation in orchards (Fernández, 2017; Steppe et al., 2008) and such knowledge of plant water status has improved understanding of forest responses to drought (Anderegg et al., 2018; Konings et al., 2017). However, we lack understanding on how plant water status and response to soil water potential gradients affect source water partitioning.

Here, we present a new controlled experiment to explore the controls of plant water status on source water apportionment and plant water update. We conduct measurements of plant and soil water isotopic ratios, coupled with measurements of plant hydraulics, and climate variables, soil volumetric water content and matric potential in a





large 2 m³ lysimeter planted with two willow trees *(Salix viminalis)*. We use this experimental design to explore plant water source partitioning during different periods of tree water deficit and periods of storage replenishment. We

leverage these known boundary conditions and different wetness to answer the general question: How does plant water status and the spatial distribution of fine roots affect water use partitioning? Specifically, we explore for inter-related sub-questions in our quest for new understanding in this regard:  1) What are the main drivers of plant water status? 2) Does plant water status affect plant water source partitioning? 3) What is the role of atmospheric demand and soil water matric potential on plant water source partitioning? 4) Is plant water partitioning directly proportional

to fine root distribution throughout the soil profile? Our work combines high spatial and temporal resolution measurements (as advocated by Penna et al. 2018) and acknowledges that plant water uptake is a dynamic processes (as shown by Volkmann et al. 2016; Barbeta et al. 2015).

## 2. Material and Methods

### 2.1. Lysimeter layout

Our experiment was conducted on a vegetated weighing soil lysimeter, situated on campus at École polytechnique fédérale de Lausanne (EPFL), in Switzerland (Figure 1). The experiment started on May 14th 2018 and ended on June 29th 2018 (approximately seven weeks). The lysimeter setup was initiated by Queloz et al. (2015) and consisted of a large chamber of 2.5 m deep and 1 m² base area fiberglass-polyester cylindrical tank. The initial experiment was set up in 2012, and the lysimeter was filled with soil and two willow trees were planted *(Salix*

*viminalis)*. In 2014, the original trees were cut down and allowed to regrow. The regenerated willow trees were approximately three years old by the time this experiment started. The soil was undisturbed since 2012. The soil column is 2 m deep and consisted of a mixture of equal proportions of local loamy sand and lacustrine sand from Lake Geneva. A geotextile mesh prevented it from clogging the 0.5 m bottom layer filled with gravel. The surface of the lysimeter was at ground level, located in an open grass field. The base of the lysimeter was accessed by an

underground chamber, where free drainage was quantified at the base using a tipping bucket (Casella Measurement, UK). The lysimeter sat on three load cells with a maximal load of 2.2 metric tons each (HBM, Germany) connected to a digital transducer (AD103C, HBM). The digital signal was logged to a computer (AD Panel32 software, HBM) at 20 s intervals.

### 2.2 Atmospheric and soil conditions

An automatic weather station (MeteoMADD, MADD Technologies Sàrl, Switzerland) located five meters away from the lysimeter provided air temperature (Ta), solar radiation (Rn) and relative humidity (RH) at 15 min intervals. We calculated vapour pressure deficit (VPD) based on RH and Ta records. We used frequency domain reflectometry probes (FDR; 5TM Devices Inc., USA) to monitor and record volumetric water content at four different depths (10, 25, 125, 175 cm, with 2 probes at each depth).  Soil matric potential ($\Psi_S$) was monitored at four

depths (25, 75, 125, and 175 cm) within the lysimeter using TensioMark® probes (ecoTech UmweltMeßsysteme, GmbH, Germany). Soil matric potential measurements were recorded as pF in HPa (instrument resolution of 0.01



pF) and later converted to MPa. All measurements were recorded at 15 min intervals using a CR1000 data logger (Campbell Scientific).

We artificially irrigated the lysimeter using a drip irrigations system. Irrigations were carried at night to avoid fractionation of the water isotope composition. We monitored meteorological conditions and lysimeter soil water storage to determine when to irrigate. Irrigation was avoid tree water stress conditions, and higher soil volumetric water content during initial phase. We supressed irrigation during the final period to generate drier soil conditions and to induce tree water stress.

### 2.3 Plant water status measurements

Our plant hydraulic measurements were conducted on the 3 main stems (two belonged to a single plant and root system, while the third belonged to a different plant). We monitored stem radius change using a non-invasive automatic dendrometer (Ecomatik, Germany; type small diameter dendrometers DD-S, accuracy ±1.5 µm). DD-S sensors were installed around the base of each main stem and in a location free of branches. Stem radius change was recorded every 15 min with a datalogger (DL15, Ecomatik, Germany). These measurements allowed observations of

tree diel dynamics including irreversible radial growth, and reversible shrinking and swelling of the living stem cells due to changes in water storage following method outlined in Zweifel et al., (2016).

We measured sap flow ($J_s$) rates from the three main stems with a heat balance gauge (EXO-Skin SGA19; SGA 25, SGA 25 Dynamax, Houston, TX, USA). We installed these sensors immediately above the dendrometers where there were no branches. Sap flow sensors consisted of a heater band that was constantly supplied with power.

A pair of thermocouples located in the top and bottom of the band measured temperature gradients associated with conductive heat loss during transpiration. This system used a heat balance method to estimate sap flow density following methods described in detail in Lascano et al., (2016). Sap flow measurements were taken every minute and the 15 min averages were recorded in a datalogger (CR1000 and AM16/32 multiplexer; Campbell Scientific). We selected the heat balance method because it was non-invasive and suited to small diameter stems. We also

instrumented the tree with a leaf psychrometer (PSY 1, ICT International, Australia) to provide high-resolution measurements of leaf water potential (MPa). Leaf water potential was recorded every 15 min. We rotated the equipment every 4-5 days to a new leaf within the same branch.

### 2.4 Sampling of tree water and soil water sources

We sampled the tree- and soil water sources in the lysimeter between May 16th and June 29th, 2018. We

collected samples for isotopic analysis of tightly bound water pool, mobile water pool, xylem water, and precipitation. We obtained tightly bound water by sampling bulk soil at four different depths (10, 25, 50, 80 and 150 cm, with two replicas per depth) every four days throughout the experiment. Samples from the 10 and 25 cm depth were collected through vertical cores using a 3 cm diameter auger. The deeper soil samples were collected laterally using the same auger through small ports on the side of the lysimeter tank. We collected samples progressively

inward from the same two access points. This was done to avoid creating a large number of ports at the same depth.





We removed approximately 8 cm of soil per sampling event, and discarded the first 3 cm. We discarded the first centimeters of soil to ensure that we were not sampling any enriched soil water due to any possible evaporation at the soil interface. We closed the access point with a sealed PVC pipe of same diameter to minimize possible preferential leakage, evaporation or condensation of water in the empty space. For the surface soil sampling locations 10 and 25 cm, we filled the open space with a dry soil. We also flagged the vertical sample locations to avoid future resampling of those areas. Soil samples were immediately stored in 12 ml Exetainer® vials (Labco Ltd, Lampeter, UK).

For more mobile water pool sampling, we used a system of radially-inserted ceramic porous cups combined with an automated pump to collect soil water (Queloz et al., 2015). We collected this mobile water every other day from five different depths (10, 25, 50, 100, and 150 cm) at three different locations distributed radially within the lysimeter. We vacuumed down simultaneously each porous cups in the lysimeters to 600 hPa for six hours prior to sample collection. We collected precipitation from a collector installed 2 m away from the willow. Natural precipitation was collected immediately after the event, or in the morning following the event on occasions when precipitation occurred overnight. We collected base seepage at the tipping bucket outlet every time there was outflow. The frequency of collection depended on the availability of water at the tipping bucket. After the tipping bucket, water was directed to a solenoid valve, and the volume of three tilts of a tipping bucket was collected in 12 mL plastic tubes. We immediately filtered the samples with 0.45-µm disk filters into a 2 mL vials (2mL Clear Vial, Canadian Life Science) and tightly sealed them to prevent evaporation.

We sampled willow branches for xylem isotopic analysis. We collected xylem samples each day for the first eleven days (May 23th); changing to every other day until June 10[th]; switching to every four days until June 22th; and returning to every day until the end of the experiment. We selected suberized branches with mature bark to avoid evaporative fractionation that might occur through unsuberized stems (Dawson and Ehleringer 1993), and sampled during midday when transpiration was at peak. We recorded the length, diameter and branch sampling locations in reference to the three tree main stems. Immediately after removal of a branch section, we covered the wound with silicone to minimize any possible evaporation from the exposed surface. The sampled xylem was quickly separated from bark and inner bark, chopped and stored in 12 ml Exetainer® vials. All samples from this experiment were stored in a refrigerator at 4°C.

### 2.5 Isotope analysis

Isotope analysis of collected waters were conducted at the Watershed Hydrology Lab at the University of Saskatchewan. Mobile water (water held above 600 hPa), precipitation, and seepage were analyzed using "laser spectrometer"-- isotope ratio infrared spectroscopy-- in liquid mode (LGR OA-ICOS CA, USA). Laboratory precision was ±1.0‰ and ± 0.2‰ δ2H and δ18O, respectively. We extracted water from bulk soil samples, and xylem using cryogenic vacuum distillation method, following Koeniger et al. (2011). Isotope analyses of extracted water from soil and xylem samples were performed using isotope ratio mass spectrometry to avoid possible spectral contamination in liquid or vapor mode. Laboratory precision was ± 1‰ and ± 0.2‰ δ2H and δ18O, respectively.



### 2.6 Spatial fine root distribution

At the end of the experiment, we sampled fine roots to assess root functional traits. We determined the fine root length density (RLD, cm of root per $cm^3$ of soil) (Gregory, 2006), root tissue density (RTD, g of dry root per root volume $cm^3$), and specific root length (SRL, m of root per g of dry root mass) (Ostonen et al., 2007) throughout the 2 m soil lysimeter profile. We collected three soil cores using a cylindrical soil auger (internal auger diameter of 54 mm) distributed radially around the main willow stems. We sampled every 25 cm until reaching the bottom of the lysimeter. We sieved and carefully washed soil samples individually using a 250 µm sieve to obtain fresh root samples. The fresh roots of each depth and core were stored in plastic bags at -6 ˚C until analysis. Individual samples from each depth and core were scanned on a flatbed scanner (Epson model V700) at 400 dpi. Images were analyzed on WinRHIZO software (Regent Instruments Inc.) to acquire the root length (cm) and volume ($cm^3$) among root diameters classes. The following root functional traits were calculated for fine roots (< 2mm) only. We converted root length per depth to root length density (RLD) based on the associated soil volume (572.55 $cm^3$). We obtained dry root mass (< 2mm) by oven drying the roots at 65 ˚C for 48 h. We then calculated specific fine root length (SRL) as the ratio of fine root length and fine root dry mass (m $g^{-1}$). Root tissue density (RTD) was obtained by the ratio of fine roots (< 2 mm) mass (g) per associated root volume ($cm^3$).

### 2.7 Data analysis

Data analyses and visualization were done using R, version 3.5.1 (R Development Core Team 2015).

#### 2.7.1 Calculations of tree water deficit and crown midday water potential

We calculated tree water deficit (ΔW) to obtain the overall plant water status based on the assumption that plant physiological responses change according to atmospheric and soil conditions (Dietrich et al., 2018; Drew and Downes, 2009; Zweifel et al., 2005). We followed the approach developed by Zweifel et al. (2016) to determine ΔW (Figure 2). We first observed the steam radius variation on the three stems to verify similarity in diel cycle. We used the averaged value of stem radius from the three stems because we observed similar trends between stem radial changes with larger differences related to growth. We de-trended the average stem radius change data for growth to obtain ΔW parameter. This was done by calculating the difference between the past highest stem radial record ($SR_{max}$) and current stem radial record ($SR_t$). The continuous $R_{max}$ indicates the horizontal growth line (GRO) that represented the value of the last maximum record observed in relation to stem radius record (Figure 2; green line for individual stems). We assumed no growth under periods of water deficit (Zweifel et al., 2016). Thus, zero ΔW represented periods when tree water storage compartments and cambial zone were fully hydrated and thus indicates optimal relative water content. We considered optimal water content when ΔW returned to zero values within a period of 24 hours, diurnal cycle. Tree water deficit was defined ΔW > 0 for periods longer than 24 hours.

We further investigated the relationship between ΔW and midday leaf water potential ($\Psi_{midday}$) to explore the hydraulic coupling between those two variables (Steppe, 2018). We did this by fitting a sigmoidal function (Supplementary material) (Dietrich et al., 2018; Steppe et al., 2006).





### 2.7.2 Environmental drivers of plant water status and water source analysis

Measurements of microclimate, soil matric potential, tree water deficit and sap flux were aggregated to daily values for comparison of temporal dynamics. We investigated the relationship between $\Delta W$ and $J_s$ with environmental variables using general linear models (stats, glm function in R). We used analysis of variance (ANOVA) with Tukey's HSD (honest significance difference) post-hoc test to compare distinct periods of tree water status.

To determine plant water source partitioning during distinct water deficit periods, we implemented a dual isotope mixing model using the MixSIAR Bayesian framework (Stock et al., 2018). MixSIAR integrates uncertainties related with isotopic compositions, multiple sources and discrimination factors (the reader is referred to Jackson et al., 2018 for more details regarding this model). We assumed no fractionation during root water uptake for this analysis (Ehleringer and Dawson, 1992); thus, we set concentration dependence to 0 in the MixSIAR framework

to reflect this decision. We examined specific days during each period to verify water source partitioning. We did not combine isotopic data from multiple days in source partitioning analysis so as not to mask isotopic signatures with average values from multiple days. We categorized soil water above and below field capacity and sampling methodology following Brooks et al. (2010) and Berry et al. (2017). Later, we identified three possible water sources for plant uptake: (a) "Mobile water", defined through samples of mobile collected through porous cups at a tension

of 600 hPa at all depths, when available. The 600 hPa determine the limit at water can be sampled. (2) "Shallow", defined through samples of bulk soil water taken at depths of 0 to 50 cm, and (3) "Deep", defined through samples of bulk soil water at 50 to 200 cm. Both shallow and deep represented water pools that were held under tensions below 600 hPa and obtained via cryogenic extraction of bulk soil samples. We compared results from the model with direct inference in dual isotope space.

## 3. Results

### 3.1 Tree water deficit (ΔW)

Figure 3 shows $\Delta W$ throughout the seven-week experiment. Stems showed daily fluctuations in $\Delta W$, with maximum values reaching 110 µm in a day. We defined three main periods based on tree water deficit conditions: $\Delta W = 0$ when trees fully recovered their optimal stem water content within 24 hr – 'no water deficit'; $\Delta W > 0$ when

trees did not recover their optimal water content and showed signs of water stress – 'water deficit' ; and $\Delta WD \geq 0$ when trees showed periods of 'intermittent water deficit'. The intermittent water deficit period contains equal amount of days under tree water deficit ($\Delta W > 0$), and under optimal hydraulic conditions ($\Delta W = 0$). These periods occurred during May 15th to May 31st ($\Delta W = 0$), June 1st to June 11th ($\Delta W \geq 0$), and June 14th to June 22nd ($\Delta W > 0$). Hereafter, we refer to these periods in our analysis. We note that these three water status periods were statistically

distinct (F = 29.72; p < 0.000, ANOVA) (Figure 3). We observed an increasing trend in $\Delta W$ towards the end of the experiment. The daily average of $\Delta W$ during each period were 18.8 µm, 33.3 µm, 54.8 µm, for $\Delta W = 0$, $\Delta W \geq 0$, $\Delta W > 0$, respectively, with maximum $\Delta W$ observed on June 15th (110.16 µm).





Sap flow rates were statistically higher during $\Delta W > 0$ (F = 12.89; p < 0.000, ANOVA). The daily average $J_s$ rate was 194 g/ h (± 59.3) during $\Delta W = 0$ , 217 g/ h (± 33.1) during $\Delta W \geq 0$ and 291 g/ h (± 26.4) during $\Delta W > 0$.

Sap flow ($J_s$) rates increased exponentially with an increase in $\Delta W$ throughout the entire experiment (Figure 4). The diel cycle of $J_s$ rates initiated at 5:00 h and resumed at 20:00 h during all periods (Figure 5). We observed greater amplitude of diel cycle during the last two periods, with the greatest being on $\Delta W > 0$. The diel cycle of $\Delta W$ revealed synchronicity with $J_s$ (Figure 5). We observed that during $\Delta W = 0$ and $\Delta W \geq 0$, increase in $\Delta W$ starts at 6:00 h, one hour later than $J_s$. While, during $\Delta W > 0$ period, both $J_s$ and $\Delta W$ started at 5:00 h, with no delay. We further observed

a smaller earlier peak in water deficit at 8:00 h, which is also observed in the $J_s$ diel cycle.

Leaf water potential measurements were interrupted on June 1[st], due to a sensor failure. Thus, recorded data only included the period of optimal water content ($\Delta W= 0$). During this period, midday leaf water potential ($\Psi_{midday}$) ranged from 0 to -6.16 MPa and the relationship between midday leaf water potential and $\Delta W$ represented a desorption curve. The slopes of a desorption curve describe the changes in amount of water in storage tissues

available to transpirations for a given water potential. This showed a direct relationship between crown water potential and tree water deficit (see Figure S1).

### 3.2 Soil matric potential ($\Psi_S$) controls on plant hydraulic conditions

Figure 6 shows atmospheric and soil storage data during the experiment. During the seven-week, average air temperature was lower during $\Delta W = 0$, with average of $17.8 \pm 2.27$ ˚C (mean ± SD), compared with $20.9 \pm 1.08$

and $21.9 \pm 2.36$ (mean ± SD) from $\Delta W \geq 0$ and $\Delta W > 0$, respectively. Vapour pressure deficit also increased towards the end of the experiment. Average VPD was $0.567 \pm 0.24$ kPa (mean ± SD) during $\Delta W = 0$, it increased to $0.625 \pm 0.18$ kPa (mean ± SD) during $\Delta W \geq 0$, and to $1.16 \pm 0.19$ kPa (mean ± SD) during $\Delta W > 0$. Average solar radiation (KPa) was $186 \pm 54.4$ (mean ± SD), $205 \pm 35.0$ (mean ± SD), $255 \pm 18.6$ (mean ± SD) during $\Delta W = 0$, $\Delta W \geq 0$, and $\Delta W > 0$, respectively. Precipitation during the experiment totalled 136 mm, 49% of which was recorded during $\Delta W$

$= 0$ period. Total irrigation was 466 mm of which 54% occurred during $\Delta W \geq 0$. This resulted in variable total soil water storage (mm) in the lysimeter and changes in $\Psi_S$ throughout the experiment (Figure 6).

Variability in $\Psi_S$ was observed throughout the experiment. $\Psi_S$ was also variable with depth. Values of $\Psi_S$ were $-0.035 \pm 0.008$ MPa (mean ± SD) at 25 cm, $-0.039 \pm 0.013$ MPa (mean ± SD) at 75 cm, $-0.074 \pm 0.008$ MPa (mean ± SD) at 125 cm, and $-0.0471 \pm 0.005$ MPa (mean ± SD) at 175 cm during the $\Delta W = 0$. During $\Delta W \geq 0$, $\Psi_S$ at

25 cm was relatively higher than other depths (closest to zero). Values of $\Psi_S$ ranged from, $-0.039 \pm 0.010$ MPa (mean ± SD) at 25 cm, $-0.144 \pm 0.063$ MPa (mean ± SD) at 75 cm, $-0.177 \pm 0.073$ MPa (mean ± SD) at 125 cm, and $-0.076 \pm 0.017$ (mean ± SD) at 175 cm. For the period of $\Delta W > 0$, $\Psi_S$ at 175 cm showed the highest measured soil tensions during this period. Values of $\Psi_S$ ranged from, $-0.631 \pm 0.764$ MPa (mean ± SD) at 25 cm, $-0.126 \pm 0.134$ MPa (mean ± SD) at 75 cm, $-0.112 \pm 0.062$ MPa (mean ± SD) at 125 cm, and $-0.060 \pm 0.0134$ MPa (mean ± SD) at

175 cm.

Table 1 shows the relationship between plant water status and atmospheric and soil conditions throughout the experiment. It summarizes the changes in drivers in plant water status over the identified periods in response to




changing environmental and soil water conditions. $\Delta W$ showed a strong relation to atmospheric variables during $\varDelta W = 0$, and no relationship was observed with $\Psi_S$. But during $\varDelta W \geq 0$, we observed strong explanatory power of the 25

cm $\Psi_S$ for $\Delta W$, but no relationship with other depths. During $\varDelta W > 0$ atmospheric variables showed no relationship with $\Delta W$, while $\Psi_S$ again showed the largest explanatory power, this time at 175 cm $\Psi_S$. Akaike Information Criterion (AIC) also indicated that $\Psi_S$ at 175 cm was the best predictor of changes in $\Delta W$. Soil water storage (mm) also showed a strong relation to $\Delta W$, changing throughout the experiment. In contrast to $\Delta W$, we found no relationship between $J_s$ and $\Psi_S$. During all periods, $J_s$ only responded to changes in atmospheric conditions.

### 285    3.3 Root distribution

We found roots in all depths of the lysimeter, including fine roots (< 2mm) at 200 cm. Overall, the fine root length density (RLD) showed a decline from 14.24 cm cm$^{-3}$ ± 5.81 (mean ± SD) at 0-25 cm to 6.47 cm cm$^{-3}$ ± 2.99 (mean ± SD) at 200 cm (Figure 7). Fine roots represented 99% of the total root length density (RLD) sampled in the lysimeter, of which the large majority (> 90% for all depths) were < 0.5 mm in diameter. Root tissue density (RTD)

was higher at shallow soil layers, with greatest value of 0.29 g cm$^{-3}$ ± 0.03 at 50 cm deep (Figure 7). Overall, the willow showed relatively high specific root length (SRL). Deep soil layers had longer SRL than shallow soil, with highest value of 132 g cm$^{-3}$ ± 54.99 (mean ± SD) at 125 cm deep (Figure 7).

### 3.4 Water source apportionment

Figure 8 shows relative water source contributions to the willow xylem water during our three main periods.

The days selected for analysis were within the identified $\Delta W$ periods, when all defined sources were available for analysis. During $\varDelta W = 0$, the bulk shallow soil water pool, bulk deep water pool and more mobile water pool contributed more equally to xylem isotopic composition, 26% (± 16%), 36% (± 23%) and 38% (± 22%) respectively. Bulk shallow water had the largest contribution to xylem water isotopic composition during $\varDelta W \geq 0$ (41% ± 24%). In contrast, the bulk deep water pool showed the largest contribution to transpiration during $\varDelta W > 0$ (52% ± 20%).

Results from MixSIAR corroborate with observations of sources and xylem on dual isotope space (Figure 10). Xylem water plotted closer to main source. The days used for this analysis were May 29$^{th}$ ($\varDelta W = 0$), June 10$^{th}$ ($\varDelta W \geq 0$), and June 22$^{nd}$ ($\varDelta W > 0$). May 29$^{th}$ was the day with lowest soil water storage during $\varDelta W = 0$. In June 10$^{th}$ trees were in deficit. On June 22$^{nd,}$ trees showed water deficit and was also the day with lowest soil water storage and lowest $\Psi_S$ throughout the entire experiment.

### 305    4. Discussion

#### 4.1 Plant water status appears to drive source water apportionment

Stable isotope ratios of $\delta^2H$ and $\delta^{18}O$ for xylem and soil pools, soil matric potential, and tree water deficit, all suggested that tree water source partitioning is driven by plant water status, and not by patterns of fine root distribution. Fine root length density (RLD) was greater at the surface (0- 25 cm) than other depths. This is consistent

with the widely observed exponential distribution of roots (Schenk and Jackson, 2002). But, importantly, the relative



proportion of tree water use did not follow patterns of root distribution common in exponential root extraction models (Schenk, 2008). Instead, water source partitioning was dynamic and followed patterns of resource availability. This is consistent with previous field observations where the depth distribution of water uptake has been shown to follow water availability and not root distribution during drought (Ellsworth and Sternberg 2015). These

findings also support root water uptake models that incorporate compensation mechanisms (e.g. De Jong Van Lier et al. 2008; Javaux et al. 2008), which suggests that plants can increase water uptake in wetter soil layers to compensate for uptake reduction in dryer layers—for maintaining water potentials and transpiration rates.

The willow's root system was 2 m deep and was characterized by high fine root length density (RLD) and high specific root length (SRL). Maximum rooting depth and specific root length are important to water resources

acquisition and plant survival during dry periods (Fort et al., 2017; Palta et al., 2011). Deep roots increase the volume of soil trees can exploit during drought (Markesteijn and Poorter, 2009; Poorter and Markesteijn, 2008; Slot and Poorter, 2007). The presence of fine roots at deeper depths appears to be important for the willow to maintain high transpiration rates and uptake of water in the deep layer during $\Delta W > 0$ in the lysimeter. Previous research has found that long specific root length (SRL) is positively correlated with high root length density (RLD) (Comas et al.,

2012; Hernández et al., 2010). Thus, the reduced water transport capacity of thinner roots is overcome by the increased root density. Hernandez et al. (2010) showed that the this associated increase in density makes the root system of long SRL species more efficient in transporting water than root systems of species with shother SRL.

The combination of functional traits observed for the willow suggest a fast resource acquisition strategy on the conservation-acquisition economics spectrum (Prieto et al., 2015; Reich, 2014). In our lysimeter, such high root

length density (RLD), long specific root length (SRL), and large proportion of fine roots (>99%), represents a trade-off between fast resource acquisition and short root life span. This is different to a  resource conservation strategy with higher tissue density (RTD) and longer life span, but lower specific root length (SRL) and root length density (RLD). The fast resource acquisition strategy observed for the willow is likely to be less affected by short-drought periods.  As shown elsewhere (Fort et al., 2015, 2017), species that adopt a fast resource acquisition strategy tend to

use more water than species that adopt a resource conservation strategies. Species that use a fast resource acquisition strategy are also more opportunistic regarding water use, without necessarily relying on shallow soil water reserves (Fort et al., 2017; Moreno-Gutiérrez et al., 2012). Thus, our observed high RLD and long SRL for the willow along with dynamic water uptake support a description of a system with fast acquisition and an opportunistic water use. Although the functional root traits provide the means for observed dynamic water uptake, they do not explain the

mechanisms for water partitioning.

**4.2 Tree water source partitioning: a hydraulic response to environmental demands**

Our results showed a dynamic root water uptake with changes in source contributions to transpiration that are driven by plant hydraulic response to $\Psi_S$. The depth of $\Psi_S$ with the greatest explanatory power over the $\Delta W$ during specific periods also showed the greatest contribution to root water uptake. This depth showed largest

contribution to xylem isotopic composition. This result contradicts the common steady-state assumption of tree water





use present in many ecohydrological investigations using stable isotopes of water (Penna et al., 2018). However, this work supports dynamic water use observed with higher frequency isotope sampling (Volkmann et al., 2016).

Water flow at the soil-plant-atmosphere continuum is driven by water potential gradients and is regulated by plant hydraulic conductance (Nobel 2009). The driving force of water uptake by roots is the negative gradient of

hydrostatic pressure between plants and soil ($\Delta\Psi$). The ability of plants to withdraw water from soil depends on plant water potential being lower than soil – "a 'tug-of-war' on a hydraulic rope" (Sperry et al. 1998). Using Darcy's Law, the flow of water from roots to soil (Q) is the product of the hydraulic conductivity of the tree (K) and the transport driving force ($\Delta\Psi$) [Q = -K $\Delta\Psi$] (Sperry et al. 1998). Thus, the relative proportion of water withdrawn per depth ($Q_{depth}$) observed in a specific time period is directly dependent on the water potential difference between the tree

($\Psi_{willow}$) and the soil at a specific depth ($\Psi_s$), for a fixed K.

We hypothesize that the observed larger relative contributions of depths with higher $\Psi_S$ to transpiration is a result of less hydraulic resistance and greater plant-soil driving force at that depth. During $\Delta W \geq 0$, $\Psi_S$ at 25 cm had the highest water potential relative to other depths. Thus, other depths offered larger hydraulic resistance at the soil-root interface. During $\Delta W > 0$, variability in $\Delta W$ was driven by $\Psi_S$, with no relationship to atmospheric variables.

Soil water potential at 175 cm was higher than other depths and showed the largest explanatory power over $\Delta W$ variability. Thus, the depth in the soil that offered less resistance to root water uptake—weaker in the "tug of water"—allowed larger transport of water to the transpiration stream. And, the depth with lowest matric potential offered more resistance, thus less water was energetically available to transpiration. The observed root functional traits provided the means to uptake water: the plant water status and hydraulics drove the uptake (Figure 10).

Recent work using transit time distributions (TTD) within several tree species, grown in controlled environment and under same soil conditions, suggests that distinct TTDs differences between species is a result of the different driving forces between specific-species and soil (Evaristo et al. 2019). Thus, it not only seems that hydraulic functioning will affect tree TTD, but also the source water partitioning in the soil profile could affect tree TTD, when investigating specific depth of water uptake. Similar findings to those of Evaristo et al., (2019) have been shown by

direct injections of tracer on tree stems. Gaines et al. (2016) reported that water transport velocity in neighboring species was controlled by the soil-leaf driving force. They found that species with more negative midday leaf water potential had higher transport velocities, and driving force, in comparison with species with more positive leaf water potential (Gaines et al., 2016). Thus, neighbour species may not only have transport velocity influenced by tree water status, but observations of tree water status may also help to understand why species explore different hydrological

niches within the soil profile (Silvertown et al., 2015).

### 4.3 Tree and environmental controls on tree water deficit

While we are unaware of any work using tree water deficit ($\Delta W$) or any other measures of tree water content to improve understanding of plant water source partitioning, the overall responses observed in our controlled experiment are consistent with results from field investigations of tree water deficit studies generally. Field

measurements of $\Delta W$ in *Callitris intratropica* have been shown to be strongly coupled with atmospheric variables


during the wet season (Drew et al. 2011). However, soil water availability was shown to be of primary importance in determining patters of ΔW during the dry season (Drew et al., 2011). Soil water potential has been shown elsewhere to better explain variability in ΔW (compared to VPD) in some tree species in Switzerland (Zweifel, Zimmermann, and Newbery 2005).

385       We believe that the minimum -6 MPa water potential observed during our initial period, and the increase in sap flow rates during relative increases in atmospheric demand during the last period is an indication of anysohydric water use behaviour (Klein, 2014; Martínez-Vilalta et al., 2014). This may also explain the strong relationship observed between ΔW and $\Psi_s$. Soil matric potential has been shown to affect ΔW of species differently. Species identified as more isohydric (larger stomatal control - maintain water potential at the cost of carbon gain) and have

390       ΔW less influenced by changes in soil water conditions, in comparison with species with more anysohydric behaviour (i.e. less stomatal control where gas exchange is maintained at the cost of decline in water status) (Oberhuber et al., 2015). During drought, anysohydric species allow decline in leaf water potential to declining $\Psi_{soil}$ (Klein, 2014; Meinzer et al., 2016; Tardieu and Simonneau, 1998). Thus, during drought anysohydric species rely largely on soil water storages to maintain transpiration rates and stomatal conductance.

395       We observed that tree water deficit (ΔW) was a good indicator of tree response to environmental drivers, such as atmospheric demands and soil water conditions. Observed variability in ΔW showed response to the primary limiting factors of stem hydraulic functioning. When soil water content was limiting, soil matric potential ($\Psi_s$) was the primary factor controlling ΔW variability. During wet conditions, atmospheric variables became the primary drivers. During periods where soil was the most limiting factor, the depth with higher $\Psi_s$ explained most of the variability in tree water deficit (ΔW).

Contrary to tree water deficit (ΔW), sap flow rates ($J_s$) did not show direct response to changes in soil water conditions during observed periods. Sap flow ($J_s$) was driven mainly by atmospheric conditions, and we observed increase in transpiration even under tree water deficit due to atmospheric demand. Tree water deficit (ΔW) showed a progressive response to environmental conditions, with days of accumulated deficit and provided information beyond the daily variability and current conditions compared to other types of water status

measurements in plants, such as sap flow and leaf water potential (Martinez-vilalta et al., 2019).

**4.4 Future tree water deficit research needs for source water understanding**

      Tree water deficit (ΔW) was first defined by Hinckley and Lassoie (1981) by de-trending stem radius changes for growth. Later, ΔW was found to be proportional to changes in the water content of the bark (Herzog et al. 1995), and then shown to provide key information about drought stress (Zweifel, Zimmermann, and Newbery 2005; Köcher,

Horna, and Leuschner 2013). Thus, tree water deficit (ΔW) provides a more integrative response of tree water status to environmental changes. Measures of relative tree water content (RWC), such as tree water deficit (ΔW) have been gaining importance in ecophysiological investigations (Martinez-vilalta et al., 2019; Steppe, 2018). Measures of water pools in plant tissue can provide a more comprehensive measure of tree hydraulic functioning than other metrics that emphasize flow (Martinez-vilalta et al., 2019). This measure can provide a more complete understanding

about net changes in water in plant tissues, including the role of storage, capacitance, cell volume and turgor loss (Choat et al., 2018; Meinzer et al., 2003; Salomón et al., 2017). Thus, tree water status may be a useful tool to



understand not only tree responses to environmental changes, such as drought (Anderegg et al., 2018), but also to improve our understanding of plant response to water availability and patterns of source water partitioning.

Figure 10 shows a conceptual model of tree water source partitioning in relation to tree water status. The depth distribution of soil matric potential explains the patterns of soil water partitioning in relation to tree water deficit. During periods of tree water deficit (Figure 10, b and c), the depth that offers less resistance to root water uptake shows the greater contribution to the transpiration stream. In the absence of tree water deficit, source water partitioning is uniform and covers depth distribution of roots (Figure 10 a). Thus, it is important to identify current plant water status to improve understanding regarding root water uptake. We argue that measurements of current

water status are relevant for understanding plant water source partitioning. Our observations indicate that the tree used both mobile and bulk soil water, but the relative contribution of bulk water was larger during all three periods. This corroborates previous findings of the use of bulk water (more tightly bound and under higher tension) by trees (Brooks et al., 2010). We acknowledge that the current methodological sampling of this more tightly bound water by extracting water from bulk soil samples may contain portion of mobile water depending on soil tension. However,

current methodology that allow sampling of tightly bound water alone do not yet exist.

We hypothesize that the use of more tightly bound water over mobile water by the willow tree is a consequence of the large density of fine roots of small dimeter (< 0.5 mm). These roots are likely in direct physical contact with the small pore space in the soil and will deplete the first millimeters of soil around them. Thus, the tightly bound water has larger contribution to transpiration because it is within the rhizosphere of the fine root system. Recent work

has shown that fine roots play a major role in facilitating water uptake in drying soils. Carminati el al. (2017) showed that root-hairless mutant plants had a significant decrease in water uptake and transpiration rates in drying soils, while their corresponding wild-type were able to considerably decrease matric potential at soil-root interface and maintain higher transpiration rates. Thus, fine roots likely enable trees to uptake more tightly bound water held under high tensions in small pore space by reducing the drop in matric potential at the root-soil interface.

Future investigations of plant water use should go beyond soil water status as the only guiding criteria for sampling and interpretations of tree water sources, and to start to ask questions of "how thirsty are trees"? In other words, to understand how, the combination of leaf water potential and tree water storage affect their current demands and response to the depth distribution of matric potential. A comprehensive understanding of tree water deficit along with isotopic analysis could be a fundamental step forward to understand species response to a depth distribution of

matric potential and source partitioning and understanding how trees drain the critical zone. This coupling of tree hydraulic (hydrometric) measurements to improve isotope interpretations of source water uptake is perhaps analogous to how catchment science began to couple precipitation-runoff hydrometrics with isotope tracing to improve understanding of the mechanisms generating runoff. And like the results of that coupling, we see much new potential for developments in ecohydrological process understanding.



## 5. Conclusion


We combined fine root distribution mapping, stable isotope tracing and plant water status monitoring to systematically investigate plant water source partitioning. We used tree water deficit as an integrative measure of plant hydraulics to understand tree response to atmospheric and soil conditions. Our lysimeter-based work showed that patterns of tree water source partitioning is influenced by plant water status and not by patterns of fine root

distribution. Depth distribution of soil matric potential explained patterns of soil water partitioning in periods of tree water deficit. These findings challenge the common assumption of a more steady-state and uniform tree water use in space, and provide empirical support for root water uptake models that incorporate compensation mechanisms. Further field-based investigations are needed to understand whether these findings are observed in the field where roots are in an unconstrained space environment, and for other species with more isohydric behaviour.

**Author contributions:** MFN, PB, DP, JJM and AR designed the experiment. MFN, PB, MA, and DP set up the experiment and collected the data. MFN performed the analysis and prepared the first draft of the manuscript. All the authors edited and commented on the manuscript.

**Competing interests:** The authors declare that they have no conflict of interest.


**Data availability:** The data from the study are currently available from the corresponding author upon request, and will be made available on data repository upon acceptance.

**Acknowledgments:** We thank Kim Janzen for assistance with laser and mass spec analysis. We thank the Laboratory of Ecohydrology at EPFL (ECHO/IIE/ENAC/EPFL) for assistance collecting samples, and a special

thanks to Bernard Sperandio for technical support throughout the experiment. This research is supported by the American Geophysical – Horton Research Grant 2019 awarded to Magali F. Nehemy, an NSERC CREATE in Water Security and an NSERC Discovery Grant to JJM.







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





Table 1. Results of relationship between ΔW and Js and single environmental variables. Result in bold indicates main environmental driver of investigated variable.

| Variable | Period | Statistics variable | Rn | T mean | VPD | Storage | Soil Ψ25 | Soil Ψ75 | Soil Ψ125 | Soil Ψ175 |
|---|---|---|---|---|---|---|---|---|---|---|
| ΔW | ΔW = 0 | R2-adj | **0.54** | 0.18 | 0.30 | 0.00 | 0.11 | -0.07 | -0.05 | -0.05 |
| | | p-value | **0.001** | 0.059 | 0.016 | 0.321 | 0.115 | 0.907 | 0.652 | 0.699 |
| | | AIC | **-44.241** | -34.959 | 112.890 | 118.570 | -33.667 | -30.742 | -30.967 | -30.903 |
| | ΔW ≥ 0 | R2-adj | 0.11 | 0.49 | 0.31 | 0.38 | **0.60** | -0.13 | 0.27 | 0.25 |
| | | p-value | 0.198 | 0.022 | 0.069 | 0.047 | **0.009** | 0.781 | 0.086 | 0.096 |
| | | AIC | -10.297 | -15.258 | -12.565 | -13.452 | **-17.474** | -8.118 | -12.080 | -11.833 |
| | ΔW > 0 | R2-adj | 0.35 | -0.03 | 0.08 | 0.80 | 0.85 | 0.79 | 0.75 | **0.88** |
| | | p-value | 0.056 | 0.405 | 0.236 | 0.001 | 0.000 | 0.001 | 0.002 | **0.000** |
| | | AIC | -12.609 | -8.535 | -9.520 | 61.156 | 58.926 | -22.991 | -21.120 | **-27.863** |
| Js | ΔW = 0 | R2-adj | 0.67 | 0.30 | **0.74** | -0.07 | -0.07 | -0.07 | -0.07 | 0.16 |
| | | p-value | 0.000 | 0.016 | **0.000** | 0.855 | 0.955 | 0.942 | 0.973 | 0.071 |
| | | AIC | -19.037 | -6.885 | **-22.815** | -0.045 | -0.009 | -0.012 | -0.007 | -3.864 |
| | ΔW ≥ 0 | R2-adj | 0.63 | **0.81** | 0.80 | 0.76 | -0.04 | 0.17 | -0.08 | -0.07 |
| | | p-value | 0.006 | **0.001** | 0.001 | 0.001 | 0.423 | 0.148 | 0.534 | 0.527 |
| | | AIC | -14.268 | **-20.249** | -19.741 | -18.066 | -4.952 | -6.957 | -4.600 | -4.619 |
| | ΔW > 0 | R2-adj | -0.13 | **0.40** | 0.16 | -0.14 | -0.10 | -0.02 | 0.05 | -0.08 |
| | | p-value | 0.800 | **0.041** | 0.157 | 0.872 | 0.631 | 0.387 | 0.274 | 0.547 |
| | | AIC | -9.349 | **-14.998** | -12.015 | -9.296 | -9.578 | -10.295 | -10.909 | -9.761 |







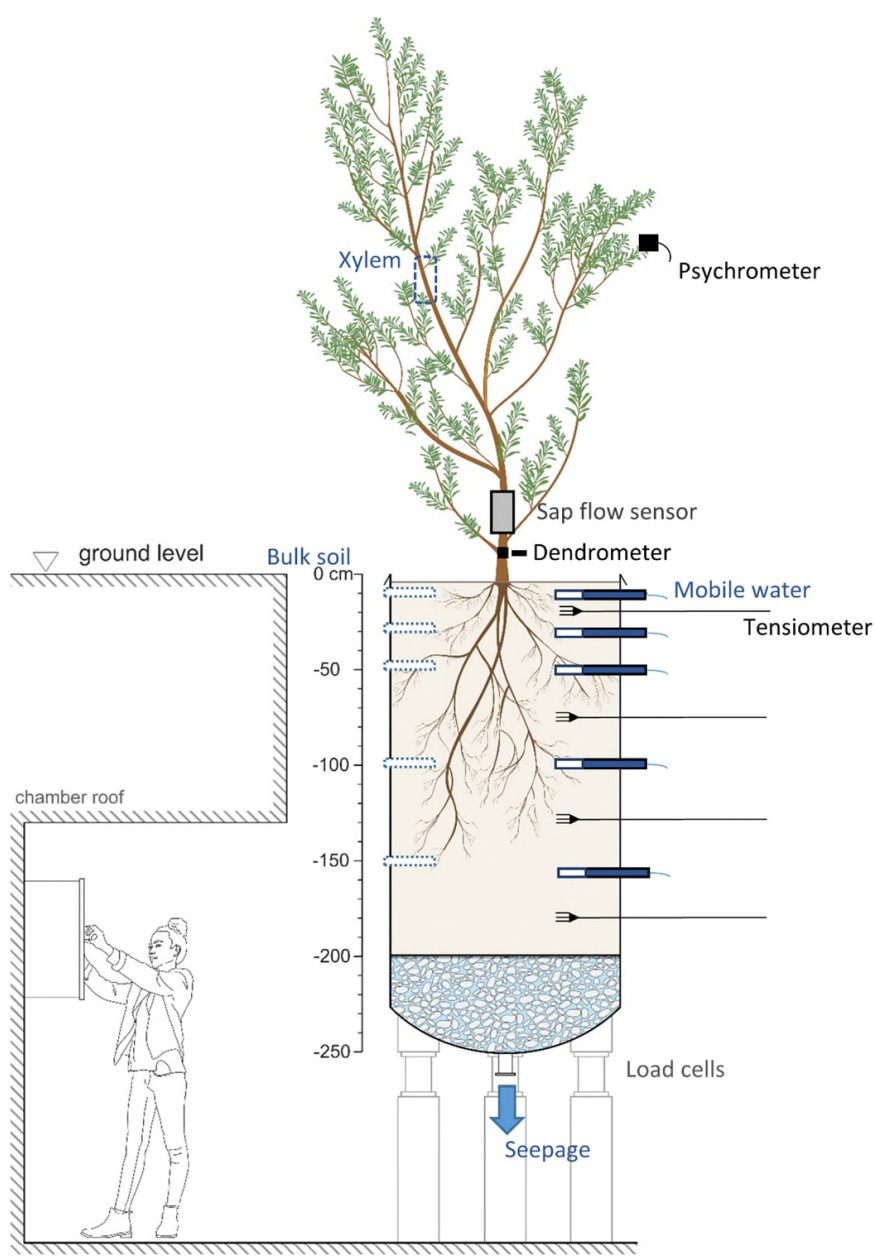

**Figure 1:** Experiment set up and design.



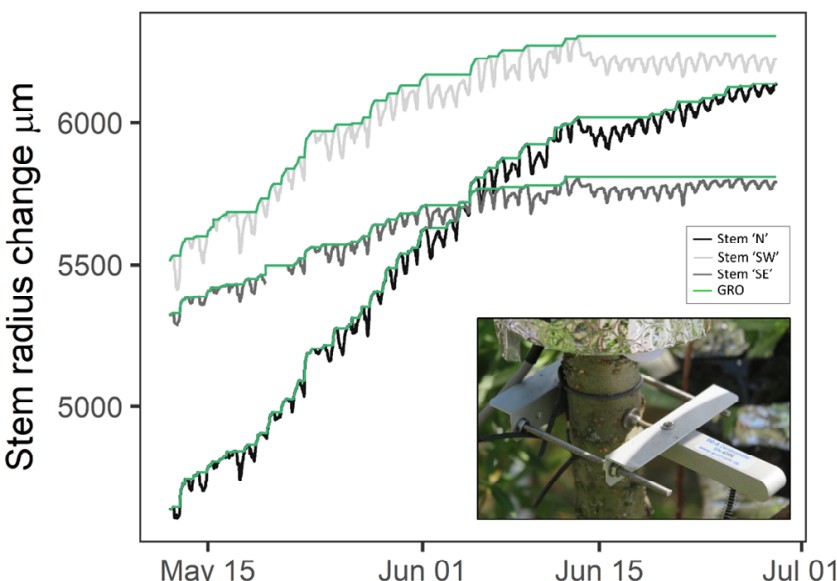

**Figure 2:** Stem radius fluctuations of tree main stems of the willow in gray tones. The difference between individual stem radius fluctuation and GRO defines ΔW. The main three stems are represented: Stem 'N' is "north" stem, Stem 'SW' is "south west" stem, and Stem 'SE' is "south east" stem. The insert shows a dendrometer used in this experiment.

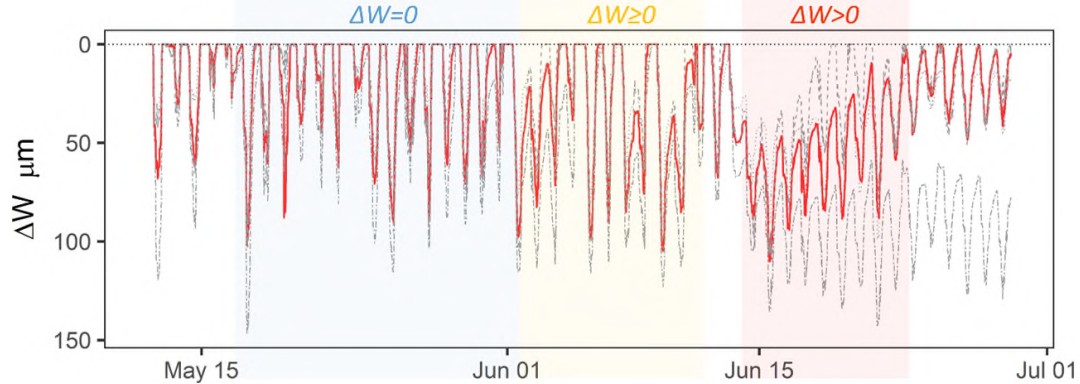

**Figure 3:** Average tree water deficit (ΔW) in red, and individual tree stems in grey. Calculations of tree water deficit (ΔW) were done following Zweifel et al. 2016. Shaded colours indicate the different ΔW periods. Blue (*ΔW=0*) indicate no water deficit period; yellow (*ΔW≥0*) indicate intermittent water deficit period; red (*ΔW>0*) indicate water deficit period.





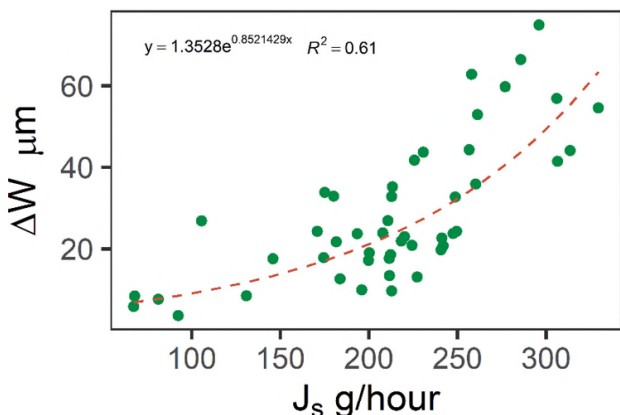

**Figure 4:** Relationship between daily tree water deficit and sap flow.

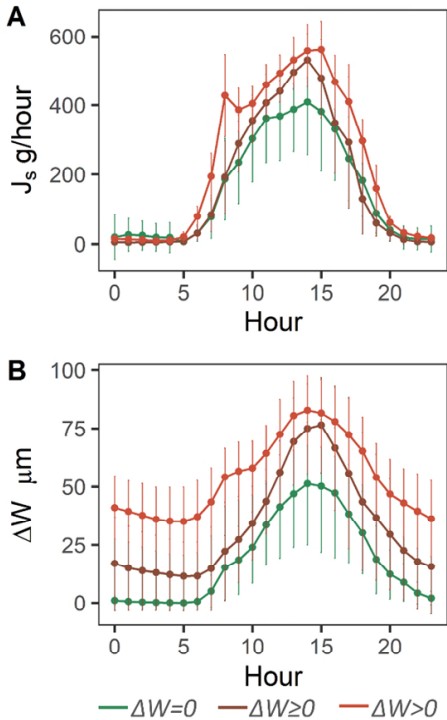

**Figure 5:** Diel cycle of hourly sap flow rates (Js) (on panel A.), and tree water deficit (ΔW) (on panel B.) averaged through the three identified periods. Periods are identified by different colours.



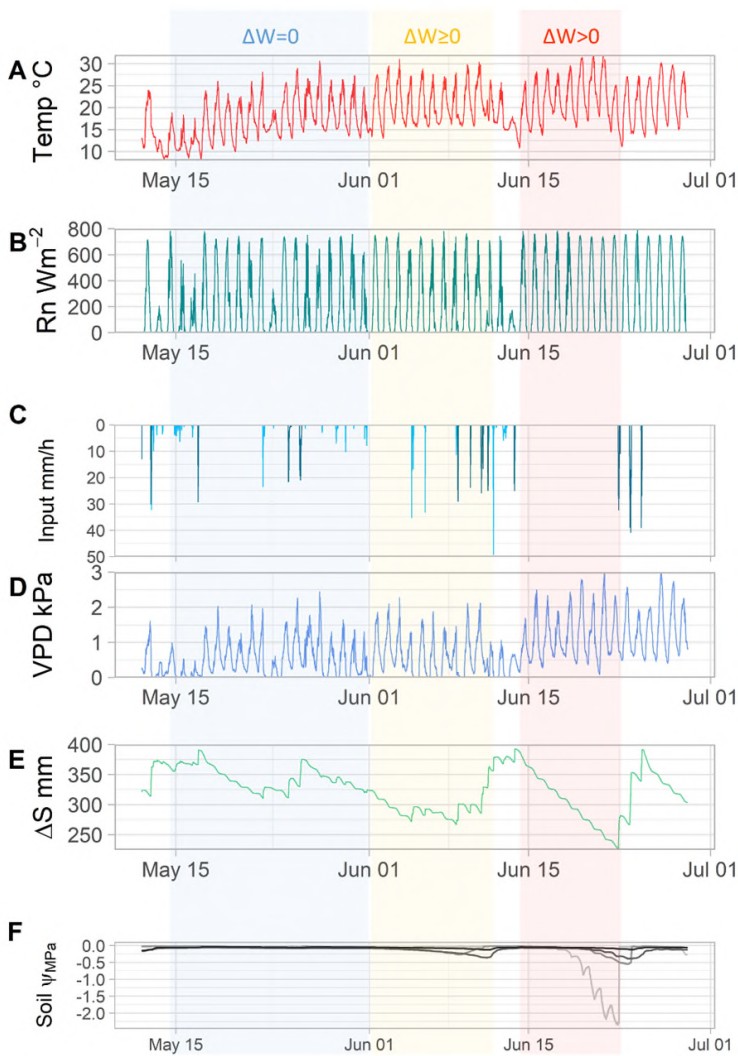

**Figure 6:** Environmental conditions during experiment. A. Temp = Temperature; B. Rn = Solar radiation; C. Input = Precipitation (light blue) and Irrigation (dark blue); D. VPD = Vapor pressure deficit; E. Soil Ψ = Soil water potential per depth (25, 75, 125 and 175 cm). Colour gradient from light gray (25 cm) to dark gray (175 cm).





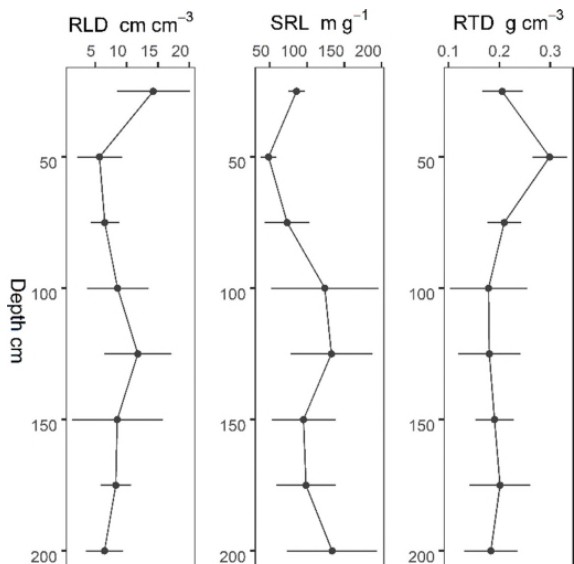

**Figure 7:** Fine root (< 2 mm) functional traits.

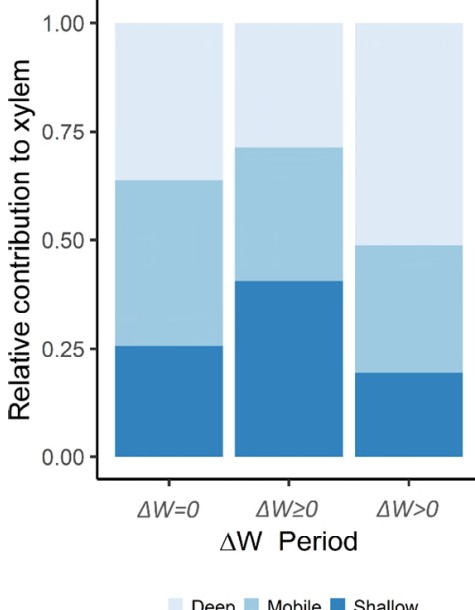

**Figure 8:** Relative water source contribution to xylem water in the three tree water deficit periods.

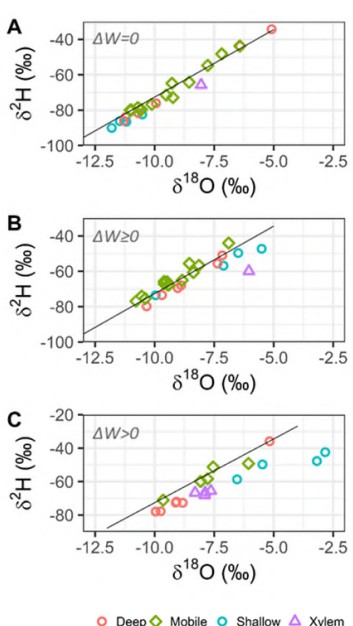

**Figure 9:** Dual-isotope plots of sources and xylem water collected in the three identified tree water deficit
periods.

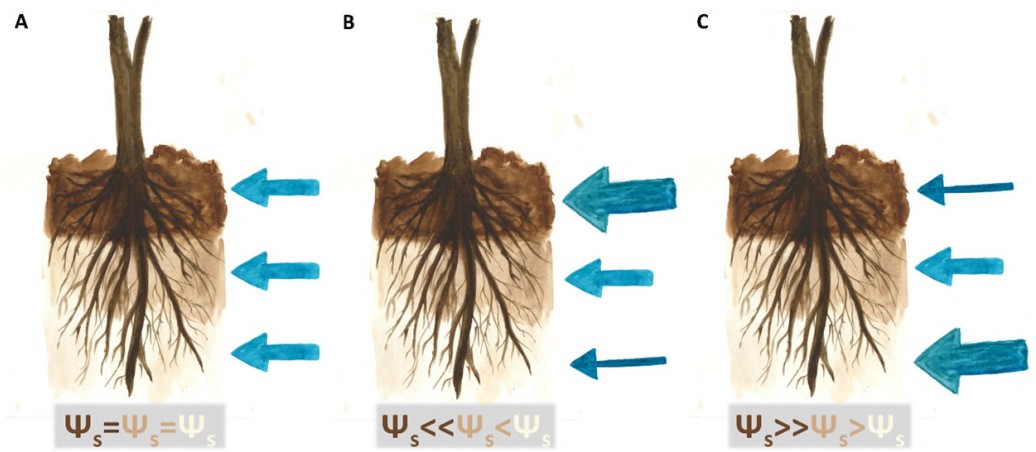

**Figure 10:** Conceptual model for tree water source partitioning. Plant water source partitioning represents
depth distribution of soil matric potential during tree water deficit periods ($\Delta W \geq 0$ and $\Delta W > 0$). Blue arrow
size represents relative contribution of specific soil depth to transpiration. Note: The period with identified
intermittent water deficit ($\Delta W \geq 0$) shows wetter conditions at the surface due to rainfall events after soil
drying out. Painting by Laura McFarlan.