# Peer review of "Analysis of $\Delta W$ and $\Psi$ midday"

_Hydrology and Earth System Sciences, 2019_

## Referee Comment (RC1) · Anonymous Referee #1 · 7 Dec 2019

This manuscript seeks to determine of tree water source changes as a function of plant water status. Specifically, the study paired point dendrometers, lysimeters, stable isotopes, and root distribution to try and discern if where plants obtained water changed as the water status of the plant and roots changed. They largely conclude that the answer to that question is "yes". Tree water source changes as a function of plant water status. I consider the question being addressed as an important one that affects our understanding of water sourcing as well as how we model water uptake across time and space. While the question is important, I do have many specific comments that I would like to see addressed.

Specific Comments

Interpretation of mixing models. The major result of this study hinges on the conclu-

sion that, using isotopes and the mixing model, water sources shift. The manuscript presents the results, with confidence intervals during W=0 and use the overlap of the confidence intervals to conclude that the values are not different. However, the results of all 3 values from the other two time periods are not reported in the text or Figure 8. Just looking at the one value reported for each of these time periods, I am concerned that there is overlap amongst the values, not allowing the data to conclude that there was a statistical shift in water sourcing across the 3 time periods. If there is no difference in these values, then there is no manuscript.

Small sample size. The data collection is thorough albeit the sample size is low. I understand that this is a tradeoff between replication and detail but it does concern me the level of conclusion being drawn from three stems on two plants. For example, does water source shift change as a function in water status in all types of plants? Plants with different rooting strategies? Plants with different ratios of fine roots? Plants with different hydraulic strategies? I am simply not confident that much can be concluded by 2 plants in 1 lysimeter.

Isotope sampling. Similar to sample size, the isotope sampling is comparing 1 single day across each of the 3 time periods. One single day on 2 plants to conclude that source water partitioning changes as a function of water status is not overly convincing that this phenomenon is consistent or real.

Issue of scale. They are comparing 3 broad periods but if water status truly affects water uptake then we would need to see analysis at finer temporal resolution. In other words, water potential (or water status) changes temporally and thus we would need to see a tighter linkages between water status and source partitioning. This would require more days where source water isotopes were sampled to derive this relationship.

Role of fine roots in explaining partitioning. The paper states "...that tree water source partitioning is driven by plant water status, and not by patterns if fine root distribution". The analysis never really provides evidence that this response is due to water availability at the expense of fine roots. In other words, the role of roots needs to be more definitively analyzed. Additionally, I am concerned about the conclusions from this that apply to field situations. This was a contained lysimeter that had fine roots throughout as opposed to broader strategies of fine root distributions seen in the field. Thus, there may be a role for fine roots that can't be captured here.

Technical Corrections Lines 76-80. Really, 3 of these questions are not effectively addressed in this paper. Lines 233-236. I do not entirely understand how the study separates out the "water deficit" and "intermittent water deficit". For example, if you looked just at June 1-3, the values and diurnal cycles look similar to June 23-24. Or June 4-6 look similar to May 17-19. In other words, the classification into 3 broad categories feels coarse and likely biases the results. Additionally, the single day sampled during "intermittent water deficit" has a diurnal cycle very similar to that sampled during "water deficit". Lines 295-300. The range of values should be reported for all 3 water sources for all three dates to determine if there is overlap. Lines 342-344, Lines 360-361. This is a bit overstated based on the data presented. Lines 385-387. This may be true, but I am not convinced that you can conclude that these are anisohydric. There are many isohydric species that have -6 water potentials.

---

## Referee Comment (RC2) · Anonymous Referee #2 · 11 Dec 2019

This study explores the role of soil and plant water status, evaporative demand and root distribution on the water use dynamics of two heavily-equipped, willow trees (Salix viminalix) installed in isolation within a buried lysimeter. Plant water deficit (noted $\Delta$W, where $\Delta$W=0 means that maximum daily stem radius has not shrunk since the previous day, and that plant water status is "optimal") is estimated from a micro-dendrometer at stem base. Soil water status is estimated from soil tensiometers at 5 soil depths, evaporative demand and tree transpiration is retrieved from lysimeter and sap flow data, and tree water origin is retrieved from water isotope (2H/1H and 18O/16O) tracing techniques and statistical mixing models, using three potential water sources: top (0-50cm) and deep (50-150cm) bulk soil water, and mobile soil water (extracted at all depths using ceramic cups at 600hPa). The authors identify three distinct periods of

plant water status (optimal, sub-optimal and transient) and find that, at least during periods when plant water status is sub-optimal ($\Delta$W>0), vertical variations in soil water matric potential, more than root distribution, explains the origin of tree water. Observing as well that soil water potential is the main explanatory variable of variations in $\Delta$W, they conclude: "plant water status drives tree source water partitioning".

First of all, I am not comfortable with the connection made between plant and soil water status. The two are linked of course but the results mostly show that plants take up soil water where it is available. The fact that the water uptake distribution may change between periods of contrasted plant water status is only because the soil water distribution also changes between these three periods.

Now, the idea that plants take up soil water where it is available is not too surprising, especially in trees where fine root length density is relatively high and well distributed across the soil horizons (Figure 7). Theories of soil water uptake by plants can explain this pattern (Cowan, 1965; Javaux et al. 2008). Even when fine root length density is not well distributed, root water uptake will depend mostly on the soil water status. This is because root water uptake increases with the soil-to-root water potential difference and decreases with the hydraulic resistance across the rhizosphere, the root endodermis and along the xylem network. In a drying soil, this network of hydraulic resistances is often dominated by the resistance through the rhizosphere, that depends on fine root density but mostly on soil hydraulic conductivity and thus soil water potential.

Also the three periods identified by the authors are quite arbitrary. They could also correspond to periods of beginning of stem growth ($\Delta$W=0), growth ($\Delta$W$\geq$0) and no growth ($\Delta$W>0), or little rain but high water content ($\Delta$W=0), more intermittent rain and (deep) soil water deficit ($\Delta$W$\geq$0) and no rain and higher (top) soil water deficit (until heavy rain comes) ($\Delta$W>0). Ideally we would want to study the relationship between plant water sources and plant (or soil) water status on a more continuous basis, a bit like the relationship found between $\Delta$W and sap flux (I guess mostly a result of an increase in both evaporative demand and functional sapwood area during periods of

positive ΔW). Having only one sampling day for each period is a bit limiting to draw definite conclusions about how plant water uptake varies over the season.

Finally, I do not understand how the different potential water sources are treated. The authors consider only three potential water sources: "mobile" (i.e. "extractable" at a suction of 600hPa) soil water at all depths, bulk "shallow" (0-50cm) soil water and bulk "deep" (50-200cm) soil water. But the bulk water includes the mobile water then. How can the authors argue: "shallow and deep [water samplings] represented water pools that were held under tensions below 600hPa"?

In conclusion, I find the experimental work carefully designed and of overall very good quality but the amount of sampling campaign for water isotope analysis is a bit limiting, the interpretation of the results is a bit problematic and the overall conclusions are mostly confirmatory.

---

## Author Comment (AC1) · 23 Jan 2020

We thank Referee #1 for reviewing our manuscript. We answer the general and specific comments below (in blue) and show how we will revise the paper based on this useful, critical input.

Anonymous Referee #1

This manuscript seeks to determine of tree water source changes as a function of plant water status. Specifically, the study paired point dendrometers, lysimeters, stable isotopes, and root distribution to try and discern if where plants obtained water changed as the water status of the plant and roots changed. They largely conclude that the answer to that question is "yes". Tree water source changes as a function of plant water status. I consider the question being addressed as an important one that affects our understanding of water sourcing as well as how we model water uptake across time and space. While the question is important, I do have many specific comments that I would like to see addressed.

Specific Comments

Interpretation of mixing models. The major result of this study hinges on the conclusion that, using isotopes and the mixing model, water sources shift. The manuscript presents the results, with confidence intervals during W=0 and use the overlap of the confidence intervals to conclude that the values are not different. However, the results of all 3 values from the other two time periods are not reported in the text or Figure 8. Just looking at the one value reported for each of these time periods, I am concerned that there is overlap amongst the values, not allowing the data to conclude that there was a statistical shift in water sourcing across the 3 time periods. If there is no difference in these values, then there is no manuscript.

We thank the Referee for pointing out this issue. We see that our previous choice of sources and the way we reported the uncertainties may have appeared ambiguous. Following comments by Referee #2, we will change our source definition by removing the mobile water end member and only focusing on the shallow and deep bulk soil water. Thus, we will provide new results (and at a higher temporal resolution) for the isotope-based source partitioning in the revised version, methods and results will reflect this change accordingly. Additionally, we will also investigate the effect of weighting the sources by the available water content at each depth.

Small sample size. The data collection is thorough albeit the sample size is low. I understand that this is a tradeoff between replication and detail but it does concern me the level of conclusion being drawn from three stems on two plants. For example, does water source shift change as a function in water status in all types of plants? Plants with different rooting strategies? Plants with different ratios of fine roots? Plants with different hydraulic strategies? I am simply not confident that much can be concluded by 2 plants in 1 lysimeter.

We appreciate the concerns raised by Referee #1 in this comment. We agree that more investigation is necessary to provide more confidence to this result, therefore we will temper the language used in our conclusions and highlight our limitations in the revised manuscript. We do want to highlight that the scientific community has been advocating for such controlled

experiments in order to advance our knowledge regarding plant water source partitioning (Penna et al., 2018). An important advantage of controlled experiments is that we are certain that the plant is not relying on other sources that we were not able to sample as it can occur in field investigations. We will discuss critically in more detail these tradeoffs and place our findings in the context of a field study, where we would not be able to sample all sources available nor capture the spatial distribution of water in the soil as can be accomplished in a lysimeter. We will address the Referee concern in the revised version by improving our sample size regarding the temporal resolution on isotopic composition of sources and xylem, and temper our conclusions to reflect what was found with our limited sample size. Further, we will propose a working hypothesis based on our results and emphasize the need to confirm such observations in field settings and across different species (L-456-459 - we will make sure we make this point clearer and earlier in the revised manuscript). We will propose a common path forward in exploring source water partitioning by incorporating measurements of tree water status, without relying only on isotopic measurements.

Small sample size. Similar to sample size, the isotope sampling is comparing 1 single day across each of the 3 time periods. One single day on 2 plants to conclude that source water partitioning changes as a function of water status is not overly convincing that this phenomenon is consistent or real.

In the original analysis, we used days where end-members were the most distinct within each period to compute xylem source water partitioning. We understand that only one day per period limits the finding of this study. We will improve the temporal resolution of isotopic sampling by including timeseries of isotopic composition of xylem and sources where more days will be shown per period during the experiment as mentioned above. We will also compute the partitioning of sources using other available days within each period. Preliminary time series visualization of isotopic composition of xylem and sources as well as new Bayesian mixing model analysis across other days show that the use of bulk water alone as end-members (see discussion with Referee #2) continues to indicate the larger use of water from deep layers during the period of water deficit. We also observed some variability in the partitioning among the different days within the same period which will be further discussed and analyzed. We will also include figures that will show the isotopic composition throughout the soil profile in relation to xylem water in the different periods to provide more information and transparency to support the analysis. We will address the changes from new analysis on the results and discussions. We will temper the manuscript's conclusion based on the sample size.

Issue of scale. They are comparing 3 broad periods but if water status truly affects water uptake then we would need to see analysis at finer temporal resolution. In other words, water potential (or water status) changes temporally and thus we would need to see a tighter linkages between water status and source partitioning. This would require more days where source water isotopes were sampled to derive this relationship.

We appreciate this fair concern of the Referee in this regard. This shows that clarity is lacking related to our definitions of 'changes in plant water status'. We will provide more detail in

explaining how we defined (and defend) the three distinct periods in the revised manuscript. We wanted to clarify that water status are periods based on continuous measurements of tree water deficit (ΔW). At the daily basis, the deficit of storage in the stem increases as transpiration rates increase and water potentials decrease, resulting in water loss from elastic tissues to offset decrease in water potentials in the xylem as observed in the literature (Zweifel et al., 2001; Steppe et al., 2006; Zweifel et al., 2005; De Swaef et al., 2015). Thus, this stem shrinkage (deficit) pattern is observed daily in trees. However, this offset in stem radius in relation to its fully hydrated state can last longer if the plant is under water stress (as defined by Zweifel et al., 2016, and observed in other field settings). Since plant water stress develops over timescales of days to weeks, understanding plant responses to changes in soil water content and atmospheric demands requires the integration of data from similar timescales. The understanding of changes in water status in our manuscript thus follows similar understanding of plant response to drought in the literature. A single measurement in time cannot provide the same understanding if not integrated over multiple days. Thus, changes in water status ('deficit' or 'no deficit', or simply what could be called 'wet' and 'dry' periods) cannot be defined daily, but whether a change in tree water deficit (ΔW) lasts for multiple days. The use of metrics that are physiological relevant and integrate different aspects of plant hydraulics (i.e. stem capacitance, water potential) in response to water availability are urgently needed to improve our mechanistic understanding in patterns of plant water use and complement stable isotopes observations to go beyond limitations imposed by this technique (i.e. uncertainty regarding fractionation process, extraction techniques, labour intensive). Tree water deficit (ΔW) (relative water content) provides this integrative physiological understanding and allows for inclusion of a temporal element to interpreting plant responses to water availability (Martinez-Vilalta et al., 2019).

We will address the Referee concern by improving the definition of periods in our revised manuscript and better reflect this choice in our discussions, along with more isotopic measurements within each period, as mentioned above.

Role of fine roots in explaining partitioning. The paper states ". . .that tree water source partitioning is driven by plant water status, and not by patterns of fine root distribution". The analysis never really provides evidence that this response is due to water availability at the expense of fine roots. In other words, the role of roots needs to be more definitively analyzed. Additionally, I am concerned about the conclusions from this that apply to field situations. This was a contained lysimeter that had fine roots throughout as opposed to broader strategies of fine root distributions seen in the field. Thus, there may be a role for fine roots that can't be captured here.

We agree with the Referee and we will remove this statement from our manuscript. We will maintain the findings on fine roots distribution in the lysimeter in the revised manuscript to support the ability of uptake throughout the observed depth, but we will remove statements that go beyond our own observations. We will address this change in the discussion and conclusion of our manuscript.

Technical Corrections

Lines 76-80. Really, 3 of these questions are not effectively addressed in this paper.

We will address this issue in the revised manuscript by narrowing the questions to the true scope of this work. This will result in changes to our overarching research question and the removal of question 3 and 4. This will also result in changes to the discussion in the revised manuscript by narrowing down to the main questions.

Lines 233-236. I do not entirely understand how the study separates out the "water deficit" and "intermittent water deficit". For example, if you looked just at June 1-3, the values and diurnal cycles look similar to June 23-24. Or June 4-6 look similar to May 17-19. In other words, the classification into 3 broad categories feels coarse and likely biases the results.

As we described above, we provide a physiological definition of tree water status, that is not based on soil measurements but directly inferred by hydrometric measurements in the plant. However, we agree that the definition of periods needs to be further clarified as it is not only a point in time but an integration of multiple days that represents plant response to soil water conditions and atmospheric demands. Therefore, the "water deficit period" (or simply 'dry period') only include days where the stem tissues are not fully water saturated and tree water deficit ($\Delta W$) does not cease completely overnight (positive values) over several days. Whereas the intermittent water deficit data (between 'wet' and 'dry') shows periods where tree water deficit ($\Delta W$) is identified but recovered shortly thereafter. For example, during June 1-3 as questioned by the Referee, we observed two days of tree water deficit ($\Delta W$) (June 1-2), but followed by a recovery day (June 3). Conversely, June 23 and 24 there is no tree water deficit ($\Delta W$), which lasted until the end of the experiment (these two days are also not part of any defined period). On June 4-6 the plant is in between days of tree water of deficit, while this is not the case for May 17-19. In the revised manuscript, we will address this issue by improving the definition of periods.

Lines 342-344, Lines 360- 361. This is a bit overstated based on the data presented.

We will temper the statements in the revised manuscript.

Lines 385-387. This may be true, but I am not convinced that you can conclude that these are anisohydric. There are many isohydric species that have -6 water potentials.

We agree with the Referee. We will remove this statement from our discussion in the revised manuscript. We will adjust the discussion accordingly.

References cited in this response

Martinez-Vilalta, J., Anderegg, W. R., Sapes, G., & Sala, A. (2019). Greater focus on water pools may improve our ability to understand and anticipate drought-induced mortality in plants. New Phytologist, 223(1), 22-32.

Penna, D., Hopp, L., Scandellari, F., Allen, S. T., Benettin, P., Beyer, M., & Dawson17, J. W. (2018). Tracing ecosystem water fluxes using hydrogen and oxygen stable isotopes: challenges and opportunities from an interdisciplinary perspective. Biogeosciences Discuss, 15(21), 6399-6415.

De Swaef, T., De Schepper, V., Vandegehuchte, M. W., & Steppe, K. (2015). Stem diameter variations as a versatile research tool in ecophysiology. Tree Physiology, 35(10), 1047-1061.

Steppe, K., De Pauw, D. J., Lemeur, R., & Vanrolleghem, P. A. (2006). A mathematical model linking tree sap flow dynamics to daily stem diameter fluctuations and radial stem growth. Tree physiology, 26(3), 257-273.

Zweifel, R., Haeni, M., Buchmann, N., & Eugster, W. (2016). Are trees able to grow in periods of stem shrinkage?. New Phytologist, 211(3), 839-849.

Zweifel, R., Item, H., & Häsler, R. (2001). Link between diurnal stem radius changes and tree water relations. Tree Physiology, 21(12-13), 869-877.

Zweifel, R., Zimmermann, L., & Newbery, D. M. (2005). Modeling tree water deficit from microclimate: an approach to quantifying drought stress. Tree physiology, 25(2), 147-156.

---

## Author Comment (AC2) · 23 Jan 2020

We thank Referee #2 for reviewing our manuscript. We answer the comments below (in blue) and show how we will revise the paper based on this thoughtful input.

This study explores the role of soil and plant water status, evaporative demand and root distribution on the water use dynamics of two heavily-equipped, willow trees (Salix viminalix) installed in isolation within a buried lysimeter. Plant water deficit (noted $\Delta W$, where $\Delta W=0$ means that maximum daily stem radius has not shrunk since the previous day, and that plant water status is "optimal") is estimated from a micro-dendrometer at stem base. Soil water status is estimated from soil tensiometers at 5 soil depths, evaporative demand and tree transpiration is retrieved from lysimeter and sap flow data, and tree water originis retrieved from water isotope ($2H/1H$ and $18O/16O$) tracing techniques and statistical mixing models, using three potential water sources: top (0- 50cm) and deep (50-150cm) bulk soil water, and mobile soil water (extracted at all depths using ceramic cups at 600hPa). The authors identify three distinct periods of plant water status (optimal, sub-optimal and transient) and find that, at least during periods when plant water status is sub-optimal ($\Delta W>0$), vertical variations in soil water matric potential, more than root distribution, explains the origin of tree water. Observing as well that soil water potential is the main explanatory variable of variations in $\Delta W$, they conclude: "plant water status drives tree source water partitioning".

First of all, I am not comfortable with the connection made between plant and soil water status. The two are linked of course but the results mostly show that plants take up soil water where it is available. The fact that the water uptake distribution may change between periods of contrasted plant water status is only because the soil water distribution also changes between these three periods.

We thank the Referee for raising this point. We would defend our approach by emphasizing that the connection between plant water status investigated with the use of dendrometers (and computed as tree water deficit) and soil water status has been previously explored in the literature in field conditions (Drew et al., 2011; Kocher et al., 2013; Oberhuber et al., 2015; Zweifel et al., 2005, 2016; Fan et al., 2019 and others). However, how this information can be used as a tool to understand plant water uptake has not been explored before. This is, in our opinion, one of the crucial point of this paper. We also want to recall that species differ markedly in the ways they adjust water balance (supply - loss) in response to environmental conditions (i.e. soil water availability). These distinct adjustments (i.e. via stomatal conductance, use of stored water – stem capacitance) reflect in different water status even when species coexist under same conditions, as shown in the literature. Therefore, plant water status has species-specific response and it is not just a function of soil conditions. If we define 'dry' or 'wet' conditions based on soil alone we are biased towards soil conditions, and we ignore species traits that drive water uptake and define extraction limits across changes in soil water potential (Sperry et al., 1998). Species-specific physiological responses are largely overlooked in ecohydrological investigations of plant water use. The use of metrics that allow

us to continually monitor physiological responses to water availability, such as tree water deficit provides opportunity to overcome limitations imposed by stable isotopes alone (i.e. uncertainty related to fractionation) and better understand species response to patterns in water availability.

Now, the idea that plants take up soil water where it is available is not too surprising, especially in trees where fine root length density is relatively high and well distributed across the soil horizons (Figure 7). Theories of soil water uptake by plants can explain this pattern (Cowan, 1965; Javaux et al. 2008). Even when fine root length density is not well distributed, root water uptake will depend mostly on the soil water status. This is because root water uptake increases with the soil-to-root water potential difference and decreases with the hydraulic resistance across the rhizosphere, the root endodermis and along the xylem network. In a drying soil, this network of hydraulic resistances is often dominated by the resistance through the rhizosphere, that depends on fine root density but mostly on soil hydraulic conductivity and thus soil water potential.

We agree that observing that plants take up soil water where it is available may not be surprising. However, we show that this uptake is dynamic and can be observed by hydrometric measurements in the plant, other than stable isotopes alone. The plasticity of plant water uptake from different soil depths has been difficult to identify in field conditions (i.e. lack of knowledge of boundary conditions) and the mechanisms remains poorly understood. We use measurements that represents plant adjustment to fluxes and water availability over time to improve mechanistic understanding in shift in water uptake given limitations in stable isotopes (i.e. uncertainties regarding to fractionation, water extraction bias, labour intensive). Our data suggests that the shift in water uptake is related to plant water stress, which we would not be able to determine by simply looking at timeseries of soil moisture at different depths. The theories and models in the literature that supports compensation mechanisms in water uptake is in our discussion (4.1 – Lines 313-317, including Javaux et al., 2008). We have also explained in our manuscript the dependency in hydraulic resistance that increases in drying soils and dependence on water potential gradient between roots and soil (Line 348-364). However, soil hydraulic conductivity cannot explain patterns of plant water uptake alone. Previous studies that have investigated shifts in water uptake among different species using stable isotopes showed that not all species change the depth of uptake with changes in soil water availability (e.g. Ellsworth and Sternberg 2015; Volkmann et al., 2016; Antunes et al., 2018). Despite similar root distribution, the distinct source water partitioning during drought conditions among species has been suggested to be associated with species-specific hydraulic traits (Volkmann et al., 2016). Plants respond differently to changes in matric potential through the soil profile (that depends for example on species hydraulic safety margins and stem capacitance). The shift in uptake is not only a metric of soil conditions, but also of the plant's ability to decrease water potential while maintaining xylem within safety margins, and hydraulic conductivity. Thus, monitoring plant water status that captures species-specific adjustments to changes in water availability using isotopes may help us understand why we see distinct patterns of plant water

use. We will change the introduction and discussion in the revised manuscript to better reflect the current understanding in the literature and address the points raised by this comment.

Also the three periods identified by the authors are quite arbitrary. They could also correspond to periods of beginning of stem growth (ΔW=0), growth (ΔW≥0) and no growth (ΔW>0), or little rain but high water content (ΔW=0), more intermittent rain and (deep) soil water deficit (ΔW≥0) and no rain and higher (top) soil water deficit (until heavy rain comes) (ΔW>0). Ideally we would want to study the relationship between plant water sources and plant (or soil) water status on a more continuous basis, a bit like the relationship found between ΔW and sap flux (I guess mostly a result of an increase in both evaporative demand and functional sapwood area during periods of positive ΔW). Having only one sampling day for each period is a bit limiting to draw definite conclusions about how plant water uptake varies over the season.

We respectfully disagree that the three periods are arbitrary, as they are based on methods described in the literature to compute tree water deficit (ΔW). Tree water deficit (ΔW) over multiple days provides information on plant water status in response to environmental drivers. Despite the daily variability in stem shrinkage, the offset in stem radius in relation to its fully hydrated state can last longer if the plant is under water stress, as defined by Zweifel et al., 2016. Thus, multiple consecutive days of water deficit indicate periods of water stress (which we define as 'water deficit' ΔW>0; or what could be defined simply 'dry'). Conversely, 'no deficit' (ΔW=0; or simply 'wet') periods are defined as periods when we did not observe any tree water deficit, meaning that tree water relations are balanced overnight and stem returns to fully saturated stages over many days. The intermediate state, where equal days of deficit and no deficit occur intermittently, is defined as the "intermittent" water deficit period.

We agree that plant hydraulics and growth are not disconnected processes, and there is evidence in the literature that shows trees will not grow under periods of water stress because of the tight relationship between xylem water potential and cambial cell expansion (Boyer and Silk, 2004; Steppe et al., 2006; Korner et al., 2015). The method we used to define tree water deficit also supports this evidence. However, we do not think that other factors can explain the observed ΔW patterns. The possible alternative suggested by the Referee that ΔW dynamics could be related to *"beginning of stem growth (ΔW=0), growth (ΔW≥0) and no growth (ΔW>0)"* is inaccurate. Despite the fact that growth (GRO, as defined in the manuscript) is tightly linked to the definition of ΔW, this interpretation does not explain the observed shrinkage in stem diameter in relation to stem fully saturated stage (water-related shrinkage) during 'no growth'. Second, growth patterns alone do not explain the variability observed in the intermittent period, as we observed stem diameter swelling related to changes in water content.  As per the alternative *"little rain but high water content (ΔW=0), more intermittent rain and (deep) soil water deficit (ΔW≥0) and no rain and higher (top) soil water deficit (until heavy rain comes) (ΔW>0)"*, our soil tension data shows that there is not a simple and unequivocal relationship between moisture and plant water status. Alternative definitions do not have a clear physiological interpretation regarding water stress as the method applied here.

We do acknowledge lack of clarity in our methodology and discussion. Thus, we will incorporate the points discussed here in our revised manuscript. We will also increase the number of sampling days observed per period as required by both referees.

Finally, I do not understand how the different potential water sources are treated. The authors consider only three potential water sources: "mobile" (i.e. "extractable" at a suction of 600hPa) soil water at all depths, bulk "shallow" (0-50cm) soil water and bulk "deep" (50-200cm) soil water. But the bulk water includes the mobile water then. How can the authors argue: "shallow and deep [water samplings] represented water pools that were held under tensions below 600hPa"?

We agree with the reviewer, the current definition of sources is limiting. We had selected sources based on the sampling methodology and statistical difference among end members that were identified using non-parametric Kruskal-Wallis test (not reported). However, during wet periods the mobile and bulk water isotopic signatures did not differ statistically. To address this issue, we will focus on bulk soil water samples and will focus on shallow vs deep sources. Time series visualization of isotopic composition of xylem and sources and preliminary Bayesian mixing model analysis show that the use of bulk water as end-members (shallow and deep) continues to indicate the larger use of water from deep layers during the period of water deficit. We also observed some variability in the partitioning among the different days within the same period which will further be discussed. We will include the isotopic composition of the soil profile in relation to xylem water to provide more evidence and understanding of the patterns observed.

In conclusion, I find the experimental work carefully designed and of overall very good quality but the amount of sampling campaign for water isotope analysis is a bit limiting, the interpretation of the results is a bit problematic and the overall conclusions are mostly confirmatory.

We thank Referee #2 for the positive evaluation of our experimental work. We agree that the current manuscript is somewhat limited by the small sample size. We will revise our results based on all the available raw data (rather than the summary findings as presented in the original draft). We will conduct new analyses based on the updated definition of water sources and address Referee #1 concerns about potential overlap in signatures of the two different water pools. Here again, we will let the data and analysis speak—and not be so definitive in our statements about what exactly is going on—but rather, discuss how different processes may be occurring based on these different perspectives discussed above and uncertainties of our data set. We will therefore tone down the definitiveness of our findings and explore openly other possible explanations, and do our best to defend which one we think is consistent with other findings. However, we do maintain our opinion that our results and conclusions are not confirmatory. Our study goes beyond what has been reported or investigated previously in the ecohydrology (plant water source investigations) literature. We use knowledge of plant water status (current physiological response to environmental conditions) simultaneously with stable

isotopes to investigate patterns of plant water use. There is a gap in the current mechanistic understanding of what drives the partitioning of sources. We show that incorporating measurements that have a physiological understanding of plant response to environmental conditions at time of uptake could fill part of this gap. Despite the broad application of stable isotopic data to estimate root water uptake strategies, several aspects of this methodology are limiting (i.e. uncertainty with fractionation processes, low sample size, labor intensive). Thus, a combination of approaches is crucial. We will also do a better job at clarifying how our work advances the logical progression in ecohydrological separation literature—and therefore, is an important and useful contribution in moving this body of knowledge forward. Measures of plant water status that uses relative water content (i.e. tree water deficit) provides an integrative approach on the understanding of plant response to drought and incorporates temporal response to changes in environmental conditions (Martinez-Vilalta et al., 2019).

**References cited in this response**

Antunes, C., Díaz-Barradas, M. C., Zunzunegui, M., Vieira, S., & Máguas, C. (2018). Water source partitioning among plant functional types in a semi-arid dune ecosystem. Journal of Vegetation Science, 29(4), 671-683.

Boyer, J. S., & Silk, W. K. (2004). Hydraulics of plant growth. Functional plant biology, 31(8), 761-773.

Drew, D. M., Richards, A. E., Downes, G. M., Cook, G. D., & Baker, P. (2011). The development of seasonal tree water deficit in Callitris intratropica. Tree physiology, 31(9), 953-964.

Ellsworth, P. Z., & Sternberg, L. S. (2015). Seasonal water use by deciduous and evergreen woody species in a scrub community is based on water availability and root distribution. Ecohydrology, 8(4), 538-551.

Köcher, P., Horna, V., & Leuschner, C. (2013). Stem water storage in five coexisting temperate broad-leaved tree species: significance, temporal dynamics and dependence on tree functional traits. Tree physiology, 33(8), 817-832.

Körner, C. (2015). Paradigm shift in plant growth control. Current opinion in plant biology, 25, 107-114.

Martinez-Vilalta, J., Anderegg, W. R., Sapes, G., & Sala, A. (2019). Greater focus on water pools may improve our ability to understand and anticipate drought-induced mortality in plants. New Phytologist, 223(1), 22-32.

Oberhuber, W., Kofler, W., Schuster, R., & Wieser, G. (2015). Environmental effects on stem water deficit in co-occurring conifers exposed to soil dryness. International journal of biometeorology, 59(4), 417-426.

Sperry, J. S., Adler, F. R., Campbell, G. S., & Comstock, J. P. (1998). Limitation of plant water use by rhizosphere and xylem conductance: results from a model. Plant, Cell & Environment, 21(4), 347-359.

Steppe, K., De Pauw, D. J., Lemeur, R., & Vanrolleghem, P. A. (2006). A mathematical model linking tree sap flow dynamics to daily stem diameter fluctuations and radial stem growth. Tree physiology, 26(3), 257-273.

Volkmann, T. H., Haberer, K., Gessler, A., & Weiler, M. (2016). High-resolution isotope measurements resolve rapid ecohydrological dynamics at the soil–plant interface. New Phytologist, 210(3), 839-849.

Zweifel, R., Zimmermann, L., & Newbery, D. M. (2005). Modeling tree water deficit from microclimate: an approach to quantifying drought stress. Tree physiology, 25(2), 147-156.

Zweifel, R., Haeni, M., Buchmann, N., & Eugster, W. (2016). Are trees able to grow in periods of stem shrinkage?. New Phytologist, 211(3), 839-849.